# Development and Immunoprotection of Bacterium-like Particle Vaccine against Infectious Bronchitis in Chickens

**DOI:** 10.3390/vaccines11081292

**Published:** 2023-07-28

**Authors:** Pengju Zhang, Tiantian Yang, Yixue Sun, Haiying Qiao, Nianzhi Hu, Xintao Li, Weixia Wang, Lichun Zhang, Yanlong Cong

**Affiliations:** 1Institute of Animal Biotechnology, Jilin Academy of Agricultural Sciences, Changchun 130033, China; pengju213@126.com (P.Z.);; 2Laboratory of Infectious Diseases, College of Veterinary Medicine, Key Laboratory of Zoonosis Research, Ministry of Education, Jilin University, Changchun 130062, China; 3Department of Policies and Regulations, Changchun University, Changchun 130022, China

**Keywords:** infectious bronchitis virus, S1 subunit, bacterium-like particles, immunoprotective

## Abstract

Infectious bronchitis (IB) is a major threat to the global poultry industry. Despite the availability of commercial vaccines, the IB epidemic has not been effectively controlled. The exploration of novel IBV vaccines may provide a new way to prevent and control IB. In this study, BLP-S1, a bacterium-like particle displaying the S1 subunit of infectious bronchitis virus (IBV), was constructed using the GEM-PA surface display system. The immunoprotective efficacy results showed that BLP-S1 can effectively induce specific IgG and sIgA immune responses, providing a protection rate of 90% against IBV infection in 14-day-old commercial chickens. These results suggest that BLP-S1 has potential for the development of novel vaccines with good immunogenicity and immunoprotection.

## 1. Introduction

Infectious bronchitis (IB) is an acute, highly contagious disease caused by the infectious bronchitis virus (IBV) of the genus *Gammacoronavirus*, family *Coronaviridae* [1]. Infected chickens initially show respiratory pathology with clinical signs of “bronchial obstruction” such as wheezing, coughing, and tracheal rales [2]. However, some IBV strains have different tissue and organ tropisms and can spread from the respiratory tract to the oviduct, kidney, and intestine, resulting in reduced egg production and the discharge of white, loose feces [3]. The disease has resulted in huge economic losses to the global poultry industry, especially in large and intensive poultry-producing countries such as China, the United States, and Brazil [4,5].

At present, the main problem making IB difficult to control is the emergence of numerous genotypes and serotypes due to mutations, deletions, insertions, and recombination between IBV genomes [4,6,7,8]. Based on the evolutionary analysis of the full-length IBV S1 sequence, it has been classified into seven genotypes, GI–GVII [6,9]. Vaccination remains the most cost-effective and efficient measure to reduce IB losses. However, the poor cross-protection between different genotypes and serotypes poses a great challenge to the prevention and control of IB [10,11]. The most widely used live attenuated and inactivated vaccines worldwide are derived from the GI genotype of IBV strains. The common H120 and 4/91 are used in China, along with the newly introduced LDT3-A and QXL87 live vaccines [12,13]. In the European Union, the Ma5 and 4/91 vaccine strains are used simultaneously or alternatively to provide protection against homologous or heterologous IBV strains [14]. However, the prevalence and potential risks of IBV mutants derived from live vaccine strains have attracted attention in recent years [4,15,16,17,18,19,20,21]. As for the inactivated IBV vaccines, their use alone generally elicits limited immune responses that are insufficient to protect chickens against challenge with virulent IBV strains [22,23]. In recent decades, several vaccine platforms have been designed to develop IBV vaccines, such as reverse genetic vaccines, viral vector vaccines, bacterial vector vaccines, DNA vaccines, subunit vaccines, and VLP vaccines [24], although these are currently only in the stage of laboratory research.

In this new era, there is still a need to develop safe, effective, and environmentally friendly IBV vaccines using other vaccine platforms. The bacterium-like particle (BLP) is a novel surface delivery system that is derived from *Lactobacillus lactis*. The system consists of a Gram-positive enhancer matrix (GEM) and an anchor protein (PA), which are the carrier and transport proteins, respectively [25,26]. GEM production is a simple process that involves the thermal acid treatment of lactic acid bacteria to remove proteins and nucleic acids, leaving behind the hollow particles of the cell wall peptidoglycan skeleton. In addition, GEM is immunostimulatory, and its peptidoglycan can effectively activate the innate immune response through TLR2-mediated signaling [27,28]. In this study, we utilized the GEM-PA system to display the S1 protein of IBV. After intranasal immunization of chickens with the constructed BLP-S1, the results showed that it was effective in preventing IBV infection, demonstrating the potential of BLP vaccines to activate the immune system through mucosal immunization.

## 2. Materials and Methods

### 2.1. Construction of Recombinant Baculovirus

The ORF (20,314–21,955) of the S1 gene of IBV strain H120 (GenBank No.: ON350836.1) was fused to the PA gene of *Lactococcus lactis* (GenBank No.: AM40667.1) at the 3’ end. The fused gene S1-PA was synthesized at Wuhan GeneCreate Bioengineering Co. and then cloned into the baculovirus expression vector, pFastBac1 (Thermo Fisher Scientific, Waltham, MA, USA), using T4 DNA ligase. The pFastBac1-S1-PA was transformed into *E.coli* DH10Bac (Invitrogen, Waltham, MA, USA), and the resulting recombinant bacmid, rBacmid-S1-PA, was identified by PCR after blue–white screening. *Spodoptera frugiperda* (Sf9) insect cells (ATCC, CRL-1711) were grown at 27 °C in SF900III insect cell medium (Gibco, Waltham, MA, USA) containing 5% fetal bovine serum (Gibco, USA). The recombinant bacmid (1 μg) was mixed with 8 μL of Cellfectin™ II transfection reagent (Invitrogen, USA) in 100 μL of SF900III medium. The mixture was incubated at room temperature (RT) for 30 min and then slowly added dropwise to a cell culture dish containing 80% Sf9 insect cells, which were then incubated at 27 °C for 96 h to obtain the first generation of recombinant baculovirus. Subsequent generations were passed through to the fourth generation. The fourth generation of recombinant baculovirus in the cell supernatant was harvested for Western blotting. Briefly, H120 vaccine-immunized chicken serum as the primary antibody (1:1000) was incubated overnight at 4 °C. After washing three times with PBST, HRP-conjugated goat anti-chicken polyclonal antibody (1:5000) (TransGen, Beijing, China) was incubated for 1 h at RT. The PVDF membrane was visualized by ECL color development.

### 2.2. Preparation of GEM

*Lactococcus lactis* strain MG1363 (MoBiTec Molecular Biotechnology, Goettingen, Germany) was cultured overnight at 30 °C in M17 medium. The harvested bacterium was then diluted 500-fold with M17 medium and incubated at 30 °C on a shaker at 180 rpm until the OD_600nm_ of the bacterial solution reached between 0.4 and 0.6. After discarding the supernatant, the bacterial precipitate was resuspended in 10 mM PBS and then centrifuged again under the same conditions. After three washes, the bacteria were resuspended in 10% trichloroacetic acid (Sigma-Aldrich, St. Louis, MI, USA) and boiled for 30 min. The mixture was centrifuged at 7000 rpm for 10 min at RT to remove the acid. The bacterium was then washed five times with PBS and counted at 2.5 × 10^9^ as 1 U. The sample was then observed under a transmission electron microscope (TEM). In brief, 20 μL of the sample was dropped onto the glowing copper grid and incubated at RT for 2 min. After allowing the copper grid to air dry, the front of the copper grid was stained with 2% phosphotungstic acid at RT for 1 min and observed under a TEM at 80 kV and 40,000× magnification.

### 2.3. Production of BLP Displaying the S1 Protein

The fourth generation of recombinant baculovirus was inoculated into Sf9 insect cells. After 5 days of incubation at 180 rpm and 27 °C, the cells were harvested by centrifugation at 4000 rpm for 30 min at 4 °C. The precipitate resuspended in PBS was sonicated and centrifuged at 2000 rpm for 15 min at 4 °C. The supernatant containing the S1 protein was quantified using a BCA protein quantification kit (Sigma-Aldrich, China). Then, 1 U of GEM was resuspended with different concentrations of S1 protein. The mixture was then incubated at 120 rpm for 2 h at RT and centrifuged at 2000 rpm for 15 min at 4 °C. The precipitate was washed with PBS and subjected to SDS-PAGE and TEM.

### 2.4. Animal Immunization and Challenge

Fourteen-day-old laying chickens (Jilin Deda Co., Ltd., Jilin, China) were randomly divided into three groups, namely, the experimental group (BLP-S1 group), the positive control group (live attenuated H120 vaccine group) (H120 vaccine, Harbin Pharmaceutical Group Biological Vaccine Co., Ltd., Harbin, China), and the negative control group (PBS group). Each group of 40 chickens was labeled prior to testing: 10 for IgG detection and swab sampling and 30 for secretory IgA (sIgA) detection and organ sampling. The procedure for immunization and challenge in chickens is shown in Figure 1. Each group was intranasally immunized three times at 2-week intervals with 100 μL/each. Fourteen days after the third immunization, chickens were nasally challenged with IBV strain M41 (GenBank: DQ834384.1) at a dose of 10^3.0^ EID_50_/0.1 mL, 100 μL/each. Meanwhile, three additional groups of 20 chickens each were immunized and challenged to calculate the morbidity. The clinical signs were monitored for 14 days.

### 2.5. ELISA

At days 7, 14, 21, 28, 35, and 42 post-first immunization (dpi) and days 5, 7, 10, and 14 post-challenge (dpc), an assortment of invasive and non-invasive samples, including blood and tracheal lavage, were collected from either live or euthanized birds. Blood was collected from the wing veins of six chickens in each group and centrifuged at 3000 rpm for 5 min to obtain serum. Changes in IBV-specific IgG antibody levels after immunization and challenge were determined using the IDEXX IBV Ab ELISA kit (IDEXX, China). For simultaneous assessment of the mucosal immune response, tracheal lavages were collected from three chickens ethically sacrificed by CO_2_. In brief, the entire trachea was carefully excised and rinsed repeatedly (10 times) with the same 4 mL of cold PBS containing 0.1% bovine serum albumin and 0.1% Tween 20 into the tracheal lumen, allowing it to pass through the length of the trachea [29]. Tracheal lavage fluid was obtained by centrifugation at 5000× *g* for 15 min at 4 °C. The levels of sIgA in the tracheal lavage fluids were determined using the chicken IgA ELISA Kit (Novus Biologicals, Littleton, CO, USA).

### 2.6. qRT-PCR

At 3, 5, 7, 10, and 14 dpc, oropharyngeal and cloacal swabs were collected from eight chickens in each group. Briefly, the swabs were gently inserted into the trachea or cloacal cleft, moved 5 times along the trachea or the cleft, and then subsequently immersed in PBS containing 2 × 10^6^ U/L penicillin and 200 mg/L streptomycin. After centrifugation at 5000 g for 10 min at 4 °C, 0.2 mL of the supernatant filtered through a 0.22 μm filter was collected and inoculated into the allantoic cavity of three 9–11-day-old specific pathogen-free (SPF) embryonated chicken eggs (Beijing Merial Vital Laboratory Animal Technology Co., Ltd., Beijing, China). After incubation at 37 °C for 3 days, the allantoic fluid was aseptically collected and mixed, and viral RNA was identified by qRT-PCR. The qRT-PCR was carried out using the primers targeting the N genes of IBV, N-F (5′-TTGAAGGTAGYGGYGTTCCTGAN-3′) and N-R (5′-CAGMAACCCACACTATACCATC-3′), which were synthesized by Bioengineering Co., Ltd., Shanghai, China. The qRT-PCR involved an initial incubation at 50 °C for 2 min followed by PCR amplification using a cycling program comprising an initial denaturation at 95 °C for 2 min and 40 amplification cycles, each consisting of denaturation at 95 °C for 15 s and annealing–extension at 60 °C for 30 s. The resulting cycle threshold (Ct) values of the reactions were determined using the Applied Biosystems Step One real-time thermocyclers (Life Technologies) [30], and the transcript levels were calculated using the 2^−ΔΔCt^ method [31].

### 2.7. EID_50_

Lungs and kidneys at 5, 7, 10, and 14 dpc from three chickens used for the collection of tracheal lavage fluids were aseptically sampled to assess the viral load in the tissues. The tissues were then sectioned, crushed, and centrifuged at 5000 g for 10 min at 4 °C. The supernatants were then filtered through a 0.22 μm filter to remove bacteria. The tissue supernatant was diluted 10-fold, and 100 μL was inoculated into the allantoic cavity of 9–11-day-old SPF embryonated chicken eggs. After three days of incubation at 37 °C, the EID_50_ of the virus in the tissues was determined by the Reed–Muench method [32].

### 2.8. Statistical Analysis

Data were analyzed using GraphPad Prism 7.0 software with the two-way ANOVA method, and the *p*-values were calculated. “*” indicates *p* < 0.05, “**” indicates *p* < 0.01, “***” indicates *p* < 0.001, and “****” indicates *p* < 0.0001.

### 2.9. Ethics Statement

All chickens were immunized and challenged in accordance with the Ethical Guidelines for the Welfare of Laboratory Animals in China (GB 14925–2001). The protocols involving animal studies were approved and supervised in accordance with the relevant guidelines and regulations of the Committee on the Ethics of Animal Experiments of Jilin Academy of Agricultural Sciences (JNK20220306-01). These indicators serve as clinical endpoints for chickens when they show severe clinical signs after challenge, such as extreme weakness, inability to feed and drink on their own, and extreme reluctance to stand for 24 h. These chickens were sacrificed by CO_2_ suffocation. All remaining chickens were also sacrificed by CO_2_ suffocation at the end of this study. The sick birds and the birds from which tissue samples were collected were buried in a strictly hygienically controlled burial pit.

## 3. Results

### 3.1. Expression of the IBV S1 Protein in the Baculovirus–Insect Cell System

The first generation of recombinant baculovirus, rBV-S1-PA, was produced by transfecting Sf9 cells with rBacmid-S1-PA. As shown in Figure 2B, the Sf9 cells were characterized by pathological effects after transfection for 96 h at 27 °C, including cell enlargement, roundness, slow or even non-proliferation, detachment, and fragmentation. The Western blot of the supernatant from cells infected with the fourth generation of rBV-S1-PA showed that the target protein was approximately 90 kDa, as expected (Figure 2C).

### 3.2. Identification of BLP Displaying the IBV S1 Protein on the Surface

To prepare GEM, *Lactococcus lactis* strain MG1363 treated with hot acid was vacuum-dried and observed under TEM. As shown in Figure 3A, the treated bacterium retained its bacterial morphology with a smooth surface, and the proteins and nucleic acids were removed from the cytoplasm, indicating a successful GEM preparation. The S1-PA protein at different concentrations was then incubated with GEM at RT. After centrifugation and washing to remove unbound proteins as much as possible, the grayscale of the S1-PA protein in SDS-PAGE was analyzed using Image J software. The results demonstrated that 1 U of GEM can bind up to 130 μg of S1-PA protein (Figure 3C,D). The BLP displaying the IBV S1 protein on the surface was designated as BLP-S1. Under TEM, it was observed that BLP-S1 had an ellipsoidal shape, and its surface was no longer smooth but covered with a layer of flocculent material, indicating that the S1-PA protein was anchored to the surface of the GEM (Figure 3E). The Western blot also confirmed the binding of the S1-PA protein and GEM (Figure 3F).

### 3.3. Determination of Antibody Levels in Sera and Tracheal Lavage Fluids after Immunization with BLP-S1

To determine the immunogenicity of BLP-S1, 14-day-old chickens were immunized three times with nasal drops in two-week intervals. At 7, 14, 21, 28, 35, and 42 dpi, the ELISA results showed that both the BLP-S1 and H120 groups can significantly increase the IgG levels in sera and the sIgA levels in tracheal lavage fluids compared to the PBS group (Figure 4). Although the IgG levels before 35 dpi in the BLP-S1 group were significantly lower than those in the H120 group (*p* < 0.01~0.0001), both groups reached a similar level at 42 dpi (*p* > 0.05) (Figure 4A). However, the sIgA levels in the BLP-S1 group were lower than those in the H120 group at all time points after immunization (*p* < 0.01~0.05) (Figure 4B).

### 3.4. Detection of Antibody Levels in Sera and Tracheal Lavage Fluids after IBV Challenge

To investigate the effect of virus challenge on antibody levels in these immunized chickens, the changes in the IgG levels in sera and the sIgA levels in tracheal lavage fluids were measured at 5, 7, 10, and 14 dpc. The ELISA results showed that the levels of IgG and sIgA decreased in all immunized groups after challenge (Figure 5). Although the IgG levels at 5 dpc in the BLP-S1 group were lower than those in the H120 group only (*p* < 0.05) (Figure 5A), the sIgA levels in the BLP-S1 group remained significantly lower than those in the H120 group for 14 days after challenge (*p* < 0.01~0.05) (Figure 5B).

### 3.5. Oropharyngeal and Cloacal Shedding after IBV Challenge

To detect virus shedding in the chickens after challenge, the collected oropharyngeal and cloacal swabs at 3, 5, 7, 10, and 14 dpc were inoculated into 9–11-day-old SPF embryonated chicken eggs. Viral RNA (vRNA) extracted from the allantoic fluids after 3 days was then confirmed by qRT-PCR. As shown in Table 1, the vRNA was present in oropharyngeal and cloacal swabs at 3 to 7 dpc in the BLP-S1 group, whereas vRNA was only detected at 3 dpc in the H120 group. Compared to these two immunized groups, the shedding rate in the PBS group remained high at 87.5% at 14 dpc.

### 3.6. Virus Load in Tissues after IBV Challenge

To determine the viral load in tissues after challenge, lungs and kidneys at 5, 7, 10, and 14 dpc were aseptically harvested from three chickens in each group to determine the EID_50_. As shown in Figure 6, the virus was detected in the lungs and kidneys of BLP-S1 and H120-immunized chickens at 5 and 7 dpc, but not at 10 and 14 dpc. Compared to the BLP-S1 group, the H120 group showed more effectiveness in reducing virus titers in the lungs and kidneys.

### 3.7. Clinical Symptoms of Chickens after IBV Challenge

The clinical symptoms of chickens in each group were observed daily after challenge. At 3 dpc, six chickens in the PBS group showed characteristic symptoms such as head shaking and nasal discharge, and one chicken was sacrificed by CO_2_ suffocation because of extreme weakness and the inability to feed, drink, and stand on its own. At 5 dpc, all chickens developed respiratory disorders and nine chickens from 4 to 8 dpc were sacrificed for humane reasons due to severe signs. Necropsy revealed heavy mucus secretion in the trachea and throat of these nine chickens. Although the symptoms of the remaining ten chickens were subsequently alleviated, three chickens still exhibited tracheal rales and sticky nasal mucus at 14 dpc. In contrast, all the chickens in the H120 group showed no clinical symptoms after challenge. In the BLP-S1 group, two chickens showed mild respiratory symptoms during the first three days, and one of the two chickens showed head shaking and the inability to feed and drink independently at 4 dpc and was sacrificed by CO_2_ suffocation. The morbidity curves were plotted based on the number of sick chickens in each group. As shown in Figure 7, the protection rate of BLP-S1 can reach 90%.

## 4. Discussion

At present, the detection rate of IBV is second only to that of the H9 subtype of avian influenza virus and Newcastle disease virus [33], posing a great challenge to traditional IBV vaccines. The development of novel IBV vaccines remains one of the most important issues to focus on in veterinary practice. Mucosal vaccination is an attractive alternative to traditional intramuscular injection. Not only is it a more acceptable form of vaccination, but it also has the potential to induce local protective antibodies, sIgA, in addition to systemic antibodies [34]. In general, oculo-nasal or intranasal administration of live vaccines can induce more rapid local protection and significant cellular immune responses after uptake by head-associated lymphoid tissues and antigen-presenting cells [24]. For example, the widely used live attenuated H120 vaccine can induce higher levels of mucosal IgA, which plays a role in local protection against IBV infection [35]. Although it is generally accepted that inactivated IBV vaccines usually elicit limited mucosal immune responses and local protection, an inactivated IBV vaccine encapsulated in chitosan nanoparticles induced significantly increased mucosal immune responses and a faster switch from anti-IBV IgA to IgG isotype after oculo-nasal administration to chickens, further eliciting both humoral and cell-mediated immune responses [36]. Overall, mucosal vaccines have been one of the hotspots for vaccine development worldwide. However, there are still fewer mucosal vaccines due to the high technical requirements, emphasizing the need to seek technological breakthroughs to enrich the types of mucosal vaccines.

In recent years, the BLP surface display system has shown great potential for the development of mucosal vaccines. BLP is immunostimulatory, and its main component, peptidoglycan, can effectively activate the innate immune response through TLR2-mediated signaling [27,28]. M cells in the upper respiratory tract and the lymphocytes in the mesenteric Peyer’s patches can better capture BLP. This effectively stimulates antigen-specific helper T lymphocytes, cytotoxic T lymphocyte responses, and sIgA secretion by B cells, thereby initiating a local mucosal and systemic immune response [23]. Most importantly, BLP has a potent self-adjuvant activity. There are two applications of BLP as an adjuvant. One is the preparation of a vaccine by direct mixing of BLP with a vaccine. It has been applied to vaccine studies such as the hepatitis B vaccine, the *Streptococcus pneumoniae* vaccine, and the influenza lysate vaccine, in which BLP provided these vaccines with a high level of protection compared to the control group [37,38,39]. The other is the use of BLP as a vector. The foreign antigen is expressed in fusion with PA and anchored to the surface of BLP to form a surface display antigen. It is currently used to display protective antigens for more than dozens of bacteria, viruses, and parasites, such as *pneumococcus* [40], *Yersinia pestis* [41], Zika virus [42], Middle East respiratory syndrome virus [25], and malaria [43].

To date, the BLP platform has not been applied to IBV vaccines. In this study, we prepared a BLP displaying the S1 protein of IBV and evaluated the potential application of BLP-S1 by intranasal immunization of chickens. The results showed that BLP-S1 was effective in inducing both humoral and mucosal immune responses in 14-day-old commercial laying chickens (Figure 4). Although there was no significant difference in the serum IgG levels between the BLP-S1 and H120 groups at days 7 to 14 after challenge with the same genotype but heterozygous M41 strain (Figure 5A), the sIgA levels in tracheal lavage fluids from the H120 group were higher than those from the BLP-S1 group (Figure 5B). Nevertheless, BLP-S1 remained effective in reducing viral shedding and viral loads in organs. At 14 dpc, no virus was detected in the swabs and organs of the BLP-S1 group (Table 1, Figure 6), and the protection rate reached 90% (Figure 7). This indicates that the BLP has potential as an alternative IBV vaccine. However, there are still many shortcomings and room for further improvement in our study. For example, although there is theoretically no need to purify the S1 protein displayed by the GEM-PA system, the purified S1 with a PA tag may be able to make the immune effects of BLP-S1 more specific. Second, the immunoprotective line of antibodies induced by BLP-S1 was not measured in this study. Intensive immunization may be a waste of immunity or cause immune damage to the body. Third, it is necessary to optimize the immunization dose for BLP-S1. Additionally, if sequential immunization is used, i.e., prime immunization with a live attenuated H120 vaccine and booster immunization with BLP-S1, it may reduce the frequency of immunization on the one hand and improve the immune effect on the other. These issues should be considered in the future development of BLP vaccines for IB or other diseases.

To summarize, given the limitations of the traditional IBV vaccines, there is an urgent need to develop new types of green, safe, and efficient IBV vaccines to eradicate IB and to achieve the concept of ecological and environmentally friendly farming so as to meet the current and even future needs for the effective prevention of IB. Compared to IBV subunit vaccines produced by the baculovirus_insect cell system [44] and IBV peptide vaccines [22,24], BLP vaccines are relatively simple to construct and have the advantages of high safety, high density of antigen presentation, and an autologous adjuvant effect. Additionally, BLP is highly stable and can be stored at RT without the need for cold-chain transport, significantly reducing the cost of the vaccine [41]. Thus, BLP has great potential as an alternative to traditional vaccines against IB and other diseases. Since IBV serotypes and genotypes are globally diverse and the currently available vaccines are not cross-protective, IBV antigens of different serotypes or genotypes can be displayed simultaneously on the GEM surface, which will be one of the future directions for the development of multivalent IBV vaccines.

## 5. Conclusions

Taken together, we expressed the S1 protein of GI genotype IBV using the insect cell–baculovirus system and constructed BLP-S1 based on the GEM-PA system in this study. After three intranasal immunizations, BLP-S1 induced efficient humoral and mucosal immune responses in the commercial laying chickens and reduced viral shedding and loads in organs after challenge. Our data indicate the potential of the BLP vaccine against IBV to prime the immune system through mucosal immunization.

## Figures and Tables

**Figure 1 vaccines-11-01292-f001:**
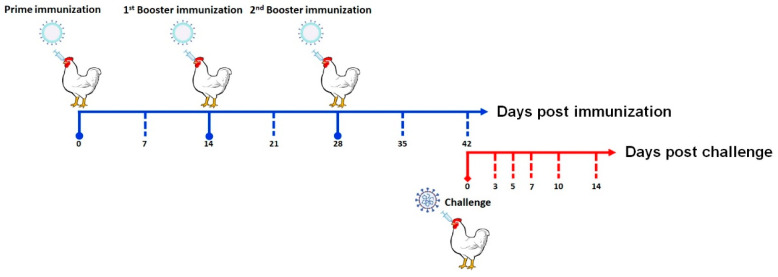
Procedure for immunization and challenge in chickens.

**Figure 2 vaccines-11-01292-f002:**
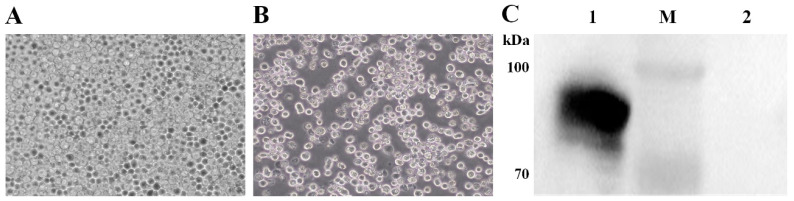
**Cytopathologic effects of recombinant baculovirus and identification of S1-PA protein by Western blot**. (Panel **A**) Sf9 cells as mock. (Panel **B**) The Sf9 cells after transfection with rBacmid-S1-PA. Magnification: 400×. (Panel **C**) Identification of S1-PA by Western blot. Lane M: protein marker; Lane 1: the S1-PA protein; Lane 2: the lysate of uninfected cells.

**Figure 3 vaccines-11-01292-f003:**
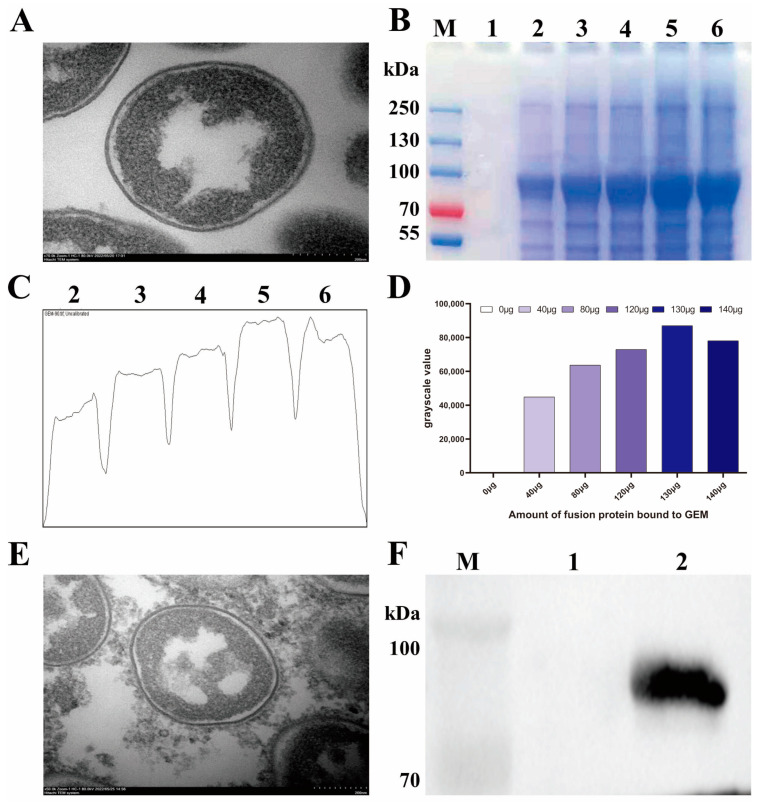
**Ability of GEM to bind to the S1-PA protein and identification of BLP-S1**. (Panel **A**) GEM particles under TEM. (Panel **B**) 1 U GEM was combined with different concentrations of S1 protein for SDS-PAGE analysis. M represents a protein marker, and Lanes 1–6 represent 0 μg, 40 μg, 80 μg, 120 μg, 130 μg, and 140 μg of S1 protein. (Panels **C**,**D**) The binding of GEM and S1-PA was analyzed and quantified in grayscale using Image J software. (Panel **E**) BLP-S1 under TEM. (Panel **F**) Identification of BLP-S1 by Western blot. M represents a protein marker, and Lanes 1 and 2 represent GEM and BLP-S1, respectively.

**Figure 4 vaccines-11-01292-f004:**
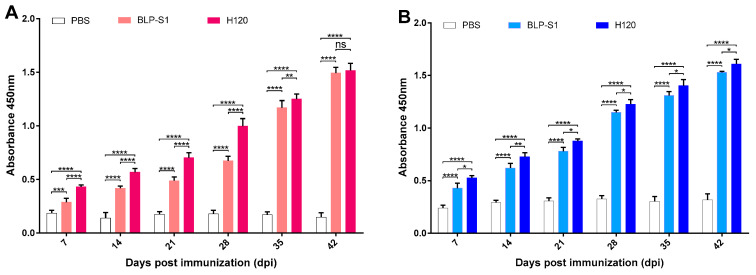
**The levels of IgG and sIgA at days 7, 14, 21, 28, 35, and 42 post-immunization (dpi).** Fourteen-day-old chickens were immunized three times with nasal drops in two-week intervals. The IBV-specific IgG in sera (Panel **A**) and the sIgA in tracheal lavage fluids (Panel **B**) were measured by ELISA. “*” indicates *p* < 0.05, “**” indicates *p* < 0.01, “***” indicates *p* < 0.001, and “****” indicates *p* < 0.0001, “ns” indicates no significant difference.

**Figure 5 vaccines-11-01292-f005:**
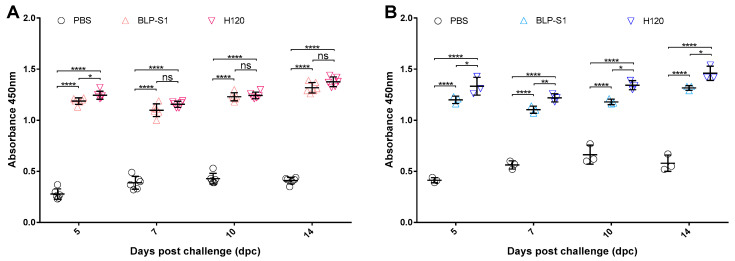
**The levels of IgG and sIgA levels at days 5, 7, 10, and 14 post-challenge (dpc).** Fourteen days after the last immunization, chickens were nasally challenged with IBV strain M41 at a dose of 10^3.0^ EID_50_/0.1 mL, 100 μL/each. The changes in IgG in sera (Panel **A**) and sIgA in tracheal lavage fluids (Panel **B**) were measured by ELISA.“*” indicates *p* < 0.05, “**” indicates *p* < 0.01, and “****” indicates *p* < 0.0001, “ns” indicates no significant difference.

**Figure 6 vaccines-11-01292-f006:**
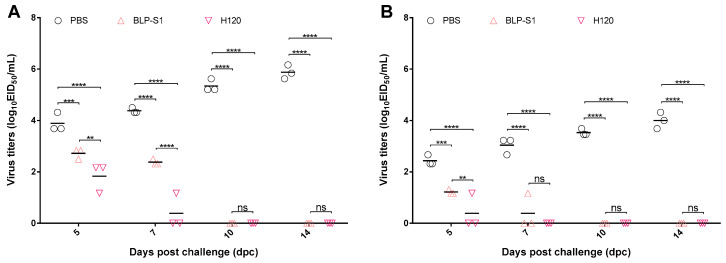
**Viral titers in the lungs and kidneys after IBV challenge.** Viral loads in the lungs (Panel **A**) and kidneys (Panel **B**) were measured by EID_50_ at 5, 7, 10, and 14 dpc. “**” indicates *p* < 0.01, “***” indicates *p* < 0.001, and “****” indicates *p* < 0.0001, “ns” indicates no significant difference.

**Figure 7 vaccines-11-01292-f007:**
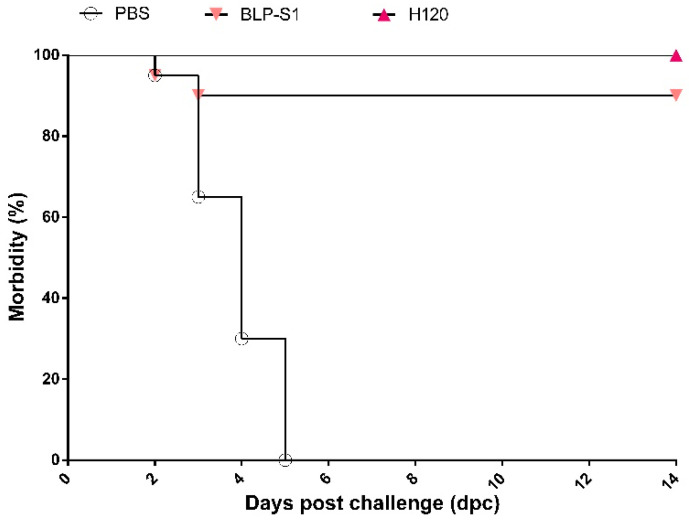
Morbidity curves of chickens within 14 days after challenge.

**Table 1 vaccines-11-01292-t001:** **The qRT-PCR detection of oropharyngeal and cloacal shedding in the 14 days after challenge.** The oropharyngeal and cloacal swabs at 3, 5, 7, 10, and 14 dpc were collected from eight chickens in each group and inoculated into 9–11-day-old SPF embryonic chicken eggs. The vRNA extracted from the allantoic fluids was then identified by qRT-PCR.

Groups	Oropharyngeal Swabs					Cloacal Swabs				
	3 d	5 d	7 d	10 d	14 d	3 d	5 d	7 d	10 d	14 d
BLP-S1	3/8	5/8	2/8	0/8	0/8	1/8	2/8	0/8	0/8	0/8
H120	1/8	0/8	0/8	0/8	0/8	0/8	0/8	0/8	0/8	0/8
PBS	8/8	8/8	8/8	8/8	7/8	5/8	5/8	7/8	6/8	7/8

## Data Availability

All the available data are provided in this paper.

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
