# Peer review of "Development and Immunoprotection of Bacterium-like Particle Vaccine against Infectious Bronchitis in Chickens"

_vaccines, 2023, doi:10.3390/vaccines11081292_

Round 1

Reviewer 1 Report

In the manuscript of  Zhang et al., the authors describe a novel vaccine against avian infectious bronchitis. A bacterial-like particles, BLP-S1, displaying the S1 protein of infectious bronchitis virus (IBV) was constructed using the GEM-PA surface display system. The vaccine was tested for immunogenicity/efficacy using 14-day-old commercial chickens. The chickens elicited IgG and sIgA immune responses after 3 intranasal immunizations, and demonstrated protection against IBV challenge.

Comments.

Line 75 – what wavelength was used for OD measurement? 600 nm?

Lines 85 and 89 – “incubation at 180 rpm” and “incubated at 120 rpm” look like mistakes.

Line 85 – change “harvested for centrifugation” to “harvested by centrifugation”.

Line 88 – what was a purity of the S1 protein? If you measured a total protein, what % S1 was in the sample?

Line 95 – What kind of H120 vaccine? Live or inactivated?

Lines 142 – 148 – It is not necessary to show these data. Construction of the bacmids can be briefly described in the Materials and methods, section 2.1, and Figure 2 can be omitted. If you decide to keep this text, the following changes have to be made: line 146 – mention expected sizes of DNA fragments; line 148 – expected size of PCR product.

Figure 3C – negative control is missing; for instance, lysate of un-infected cells.

Lines 168 – 169. It is critical to mention that unbound protein was washed.

Line 204 – It is surprising that there is no increase in the antibody titers after the challenge. Even control chickens are expected to show some titers at day 14 post-challenge.

Table 1. Looks like you show numbers of positives. How these positives were determined? What was a cut-off Ct value? This can be explained in the Materials and Methods, section 2.7.  

Line 238 – reference to Figure 8 is missing.

Lines 282 – 285. These are not conclusions from the results of the study. This can be part of discussion, but not a conclusion.

Overall, English is good. Minor edition is required. 

Reviewer 2 Report

This is an interesting manuscript that describe the methodology to generate bacterial-like particles (BPLs) which display the S1 subunit of infectious bronchitis and how these can be used to produce an immune and protective response to a viral challenge in a chicken model of infection following intranasal immunization, albeit less effective than vaccination with live attenuated vaccines (H120). They demonstrate the utility of this platform for further vaccine development. The manuscript is nicely structured and written and results clearly shown. The rationale for the sampling regime of the birds in vivo work should be expanded as well as why there is a lack of clear and concise end points for the infected birds, that were allowed to die causing unnecessary suffering.

Some further comments that the authors may want to address

Line 81 – explain the sample processing for TEM; the method described seems more suitable for SEM.

Line 105 – why were only 6 random chickens sampled in each group? Were the same animals sampled each time point? Were the birds allocated individual IDs?

Line 109 – how was the tracheal lavage performed and how severe was this procedure? Was it ante or post-mortem? Please specify at what time point it was collected and if the same birds samples for blood were also used for tracheal lavage

Line 113 – why were 8 animals swabbed (and not 6)? Were this always the same animals to evaluate shedding progression? Did birds display clinical signs at the time?

Line 116 – why was the virus passaged first in eggs and swabs not tested directly as clinical samples for PCR?

Figure 3. A and B needs higher magnification and definition, and a scale bar needs to be added (or magnification details added to figure legend)

Figure 4. A and E needs magnification detail or scale bars added

Table 1 will benefit from a better spatial separation between oropharyngeal and cloacal swabs

Clinical symptoms: further details of which birds (how many) displayed clinical signs is needed, and if those birds were sampled. The authors need to explain why no clinical end points were used in the study and chickens were allowed to succumb to infection and what ethical justification was used

Reviewer 3 Report

 Moderate editing of English language required

Round 2

Reviewer 1 Report

The manuscript has been sufficiently improved to warrant publication in Vaccines. 

Author Response

Thanks for your work again.

Reviewer 2 Report

I would like to thanks the authors for their rebuttal letter and addressing my comments which mostly they have done sufficiently.  The magnification of the TEM images seems to have been incorporated in the images, although they are not big enough to be read in the manuscript, so it would still benefit from using a scale bar or adding to the figure legend. The authors have not addressed what were the clinical end points identified as part of the ethical review of the experimental design that they would use to prevent unnecesary suffering of the challenged animals. It is ethically questionable to allow these animals to die and not identify clinical end points to cull them before they are found dead once it is clear the severity of the infection and disease will not allow a different outcome

Reviewer 3 Report

I have already reviewed the first version of this manuscript. As pointed before, the technical approach and the description of the results are the strength of the article. However, the manuscript lacked a good scientific foundation (in the Introduction) and has not been properly compared with other studies in the Discussion. Therefore, I recommended a major review for authors to prepare a more scientifically based scientific study.

The authors improved the manuscript. Some of my main suggestions were accepted and included in the manuscript. However, the text still deserves an additional revision step to make the whole text clearer and to write sentences that make sense. For example: avoid terms like "more destructive" in the last sentence of the Introduction; remove "etc." from the sentences; correct the sentence “which are produced by reference to local variant serotypes”, among others inaccuracies.

In addition, I suggest to explain the IBV genetic and antigenic diversity in the second paragraph from the Introduction. This information is necessary to understand the main vaccines currently in use. Also the Discussion need to be further substantially improved, including a mandatory paragraph about the limitations of the study, highlighting the needs of additional text in poultry farms.

Finally, a carefully revision is necessary to correct English grammar . 

It is Ok.
